# A Spiral Distributed Monitoring Method for Steel Rebar Corrosion

**DOI:** 10.3390/mi12121451

**Published:** 2021-11-26

**Authors:** Jianzhi Li, Yiyao Zhao, Junjie Wang

**Affiliations:** 1Structure Health Monitoring and Control Institute, Shijiazhuang Tiedao University, Shijiazhuang 050043, China; 2School of Materials, Shijiazhuang Tiedao University, Shijiazhuang 050043, China; zhaoyiyao1995@163.com (Y.Z.); junjiewang315@163.com (J.W.)

**Keywords:** spiral wound fiber, corrosion monitoring, Brillouin distributed sensor, reinforced concrete

## Abstract

This paper proposes a novel spiral-wound, optic-fiber sensor to monitor the corrosion of steel bars. At the same time, the winding parameters, such as winding angle and pitch, were first theoretically deduced. Then, to decrease light loss, a practically distributed sensor wound onto the protective mortar layer was developed by increasing the winding curvature radius. The spiral distributed sensors were experimentally verified for their feasibility. Experimental results showed that the spiral fiber strain depended on the thickness of the protective mortar layer. Furthermore, the spiral distributed strain well reflected the cracking process of concrete. In addition, the concrete cracking time depended on the thickness of the protective concrete layer. Accordingly, this method is feasible for evaluating the initial and final cracking behaviors of concrete structures and provides a sight for steel bar corrosion.

## 1. Introduction

Reinforced concrete structures are widely used in civil engineering applications for their durability and fire resistance. However, most reinforced concrete structures fail to meet their engineering requirements owing to insufficient durability [1]. This is mainly attributable to the damage resulting from the corrosion of steel bars, further affecting the concrete structures [2,3]. Over the past few decades, theoretical analysis of the reinforced concrete structure damage induced by corrosion [4,5,6,7,8,9,10] has shown that the corrosion process is divided into three stages: the corrosion-expansion stage, the protective-layer-cracking stage, and the crack-extension stage. Meanwhile, many techniques have been proposed and developed to monitor steel corrosion in concrete structures [11,12,13]. These are mainly divided into two categories: the first category includes some traditional electrochemical monitoring methods, such as the half-cell potential method, the constant-current method, the AC-impedance method, the electrochemical-noise method, and the concrete-resistance method [14]. The indispensable reference electrode [15,16] is vulnerable, owing to its time-varying characteristics. Therefore, steel corrosion cannot be accurately monitored at any time during long-term service. A second category comprises corrosion-monitoring methods based on optical fiber sensors. Optical-fiber sensing technology has attracted extensive attention, owing to its advantages, such as the use of lightweight fibers, high measurement accuracy, strong anti-electromagnetic interference capability, and corrosion resistance of the fibers. Consequently, it has been gradually applied to monitoring steel corrosion [17,18,19,20,21,22,23,24]. However, the corrosion sensors used in the optical fibers can only monitor corrosion at certain points and fail to measure corrosion at any location along a steel bar.

Hence, this study developed a spiral distributed fiber sensor to monitor the corrosion of steel bars in concrete structures. Compared to the existing distributed optical-fiber sensing methods, the novelty of the spiral distributed fiber lies in increasing the corrosion-mass-loss tested range and lengthening the tested distributed fiber sensor, which is attribute to the reduced light loss of the distributed sensor. Furthermore, the desired winding parameters were first achieved by theoretical analysis and experimentally verified. Thus, a spiral distributed method was utilized to measure steel corrosion in the long-term service of rebar.

## 2. Materials and Methods

### 2.1. Effect of Winding Radius on the Bending Curvature of Optical Fiber

The spiral distributed winding fiber was unrolled, as shown in Figure 1. Here, α is the helix angle, which represents the rising angle of the winding fiber, while β, called the winding angle of distributed fiber, is the complementary angle of α.

The equations of the spiral curve are expressed as follows:(1)$$\begin{array}{l} x=r\cos\theta \\ y=r\sin\theta \\ z=\pm \frac{L}{2\pi }\theta =\pm r\theta \cot\alpha \end{array}$$
where r is the radius of the steel bar (cm), *L* is the spiral distribution pitch (cm), α is the winding angle, and θ is the center angle, θ=ωt.

The dependence of the bending curvature *ρ* of the spiral distributed fiber upon the spiral winding radius and winding pitch is illustrated as follows [25]:(2)$$\rho =\frac{2\pi }{\sqrt{{\left(2\pi r\right)}^{2}+L^{2}}}$$

Simultaneously, the reciprocal of the bending curvature *ρ* is the bending radius of a spiral distributed sensor. Moreover, the bend losses of fiber depend on the bending radius. The bend losses of fiber exponentially fall with an increasing radius and ultimately keep stability [26]. Accordingly, to decrease the light loss induced by fiber bend, the minimal bending radius was achieved in this study.

Figure 2 depicts the relationship between the spiral winding radius, the spiral distribution pitch, and the curvature of the fiber. The curvature of spiral winding decreases with the increasing winding radius when the winding pitch is constant. In addition, the curvature also depends on the winding pitch of the distributed fiber. When the winding radius is unchanged, the fiber curvature gradually decreases with the increasing fiber pitch. The above-mentioned phenomena are clearly shown in Figure 3. Accordingly, the increasing fiber winding pitch can not only decrease the light loss, but also lengthen the tested steel bar.

### 2.2. Effect of Spiral Angle on a Spiral Distributed Sensor

The dependence of the spiral stress σ upon the principal stress $\sigma_{x}$ and the winding angle α of the spirally distributed fiber are expressed as:(3)$$\sigma =\sigma_{x}\cos^{2}\beta =\sigma_{x}\cos^{2}\left(\frac{\pi }{2}-\alpha \right)$$

Equation (3) indicates that the fiber stress can be decreased due to the helix angle. Thus, the distributed fiber withstands the greater corrosion expansion stress, compared with the ring distributed sensor. Therefore, the spiral method improves the corrosion test range of the steel bar.

The sensor specimen was fabricated using a carbon steel bar with a diameter of 16 mm. Based on Equation (2), the bending curvature of the helical fiber was calculated based on the different winding diameters, helix angles, and winding pitches, as shown in Table 1.

Using the Brillouin optical time-domain analysis (BOTDA), the following results were concluded, based on the above theoretical calculation and analysis:(1)The curvature of the winding fiber decreased with the increase in the helix angle and the diameter of the steel bar. At the same time, we concluded that the lower the helix angle and the spiral pitch were, the greater the curvature of the winding fiber was.(2)(When the winding diameter was 16 mm, the optical signals of the distributed sensing fiber with a 60° helix angle were obviously better than those with helix angles of 30°, 45°, and 55°; then, the light loss of the distributed sensor with a 60° helix angle was less than that of the distributed sensor with the helix angles of 30°, 45°, and 55°. The bending curvature of the fiber should preferably be less than 0.625 cm^−1^. Then, the corresponding bending radius of a spiral distributed sensor is approximately 1.6 cm. The corresponding macro-bending loss is in a lower range [26], which has basically no effect on a spiral distributed sensor.(3)The mortar layers of the steel bar were casted. Their thicknesses were 5, 8, and 10 mm, respectively. Then, the corresponding winding radii of the same helix angle distributed fiber (30°) were 26, 32, and 36 mm, respectively. Moreover, the bending curvature of the optical fiber can be calculated, as shown in Table 1. The bending curvature of less than 0.667 cm^−1^ is desirable.

In summary, the spiral method improves the corrosion rate test range of steel bar. The desired curvature of the winding fiber (0.667 cm^−1^) is a critical parameter for the distributed sensor, which contributes to not only decreasing light loss, but also to improving strain accuracy of the distributed sensor. At the same time, the fiber curvature is difficult to intuitively measure. We transformed the fiber curvature into the helix angle and spiral pitch was experimentally measured. Hence, to decrease the fiber light loss, the practically distributed sensor wound on the protective mortar layer was proposed.

## 3. Experiment 

A grouting experiment was carried out with a cement grouting gun. The mixing ratio of water, cement, fine aggregate, and water-reducing agent was 210:500:1000:3 in weight. The specimens with the above-mentioned thicknesses of the mortar layers were named 5 #, 8 #, and 10 # specimens, respectively. The schematic diagram of the distributed sensor is shown in detail in Figure 4. The three steel bar corrosion sensors were made and embedded in the concrete specimens.

Finally, the electric accelerated corrosion experiments were conducted. Before the steel bar was energized, the outer optical fiber with a spiral distribution was fixed on the mortar, as shown in Figure 5. Simultaneously, to protect the bare 2 cm steel bars at the bottom of the specimen from corrosion, the bottom surface of the mortar specimen was also sealed with AB epoxy glue. Subsequently, the mortar specimen was completely immersed in a water tank containing 5% NaCl solution and started to be corroded. There were three identical HY3005MT digital DC power suppliers used in the accelerated corrosion testing. The power suppliers have a steady current of 0.1 A. It was assumed that the corrosion of rebar is uniform. Meanwhile, the distributed fiber strains of 5 #, 8 #, and 10 # specimens were recorded by the BOTDA system. The BOTDA analyzer was a Japanese-made Neubrex 6040A with a 10 cm spatial resolution. The schematic diagram is shown in Figure 6 and Figure 7.

## 4. Results and Discussion

The relationship between fiber strain and corrosion rate is shown in Figure 8, Figure 9 and Figure 10. Obviously, the spiral fiber strain depends on the corrosion mass loss rate of the steel bar. The corrosion mass loss rate is the ratio of the corrosion mass loss to the initial rebar mass, named the corrosion rate. The spiral fiber strain increases generally with the corrosion rate. Moreover, the sensing fiber strain with the corrosion rate is divided into three stages, i.e., Stage I, Stage II, and Stage III. The findings are consistent with the references. Therefore, the spiral distributed sensor reflects well the degradation of reinforced concrete.

(1)Stage I: In this duration, the steel bar was slowly and gradually corroded. The volume of the corrosion products accumulated, leading to a subsequent increased expansion force, which correspondingly contributed to the increased fiber strain. The helix fiber strain increased almost linearly with the corrosion rate.(2)Stage II: The fiber strain increased rapidly, which was attributed to concrete cracking. In addition, the strain of the distributed fiber sensor located in the steel bar (the inner fiber) was greater than those of the distributed fiber sensor located in the mortar layer (the outer fiber). The sharp increase in fiber strain is obviously shown in Figure 8, Figure 9 and Figure 10. In this duration, AB epoxy glue on the bottom surface of the mortar specimen came off, so the AB glue was reused to protect the bare 2 cm steel bars, as shown in Figure 8.(3)Stage III: The fiber strain remained unchanged or was less than the peak strain, owing to the rust loss.

In addition, there was stochastic fluctuation characteristic of spiral fiber strain in the third stages of 5 #, 8 #, and 10 # in Figure 8, Figure 9 and Figure 10. The test data were checked afterwards. In general, a single peak of BGS spectrum shown in Figure 11 is desired, which failed to cause the measurement error. On the contrary, the double peaks in this BGS spectrum and chaotic BGS spectrum, as shown in Figure 12a,b, caused the strain fluctuation. The above-mentioned phenomena were the origins of the tested frequency shift error, which in turn led to the inaccuracy of the spiral fiber strain.

These above-mentioned fiber strain vibrations showed good agreement with the changes in the concrete structure. There were three stage of distributed fiber strain in good agreement with the three stages of the concrete structure, as shown in Figure 13. In this figure, it can be seen clearly that the corrosion products flowed from the specimen into the water due to the concrete cracks. Mechanical analysis was carried out for the concrete with cover thickness c and reinforcement diameter d. In the figure, δ represents the deformation of concrete due to rust expansion force. At first, the corrosion products filled the porous area and the elastic deformation was generated in the corrosion duration. As the corrosion rate increased, the expansion force generated by the corrosion products exceeded the tensile strength of the concrete. Subsequently, concrete cracking was generated. Hence, a sharp increase in the fiber strain occurred, owing to the generation of concrete crack, until the cracks propagated to the surface of the specimens. Therefore, the concrete cracks were generated, propagated, and penetrated the specimen, which directly led to the corrosion products’ outflow into the water through the concrete cracks. The generation and outflow of the corrosion products of steel reached a state of dynamic equilibrium or the more the rust flowed out. Hence, the spiral fiber strain may be unchanged or less than the peak strain, owing to the rust loss in the last duration.

Moreover, the cracking times of specimens 5 #, 8 #, and 10 # specimens were approximately 6, 10, and 16 h, respectively, as observed in Figure 8, Figure 9 and Figure 10, and in Table 2. At these durations, their corresponding corrosion rates were 0.16, 0.2, and 0.35%, respectively. Therefore, it was concluded that the cracking times and corresponding corrosion rates were highly related to the thickness of the protective layer. The concrete crack was well recorded, as shown in Figure 14. Obviously, the cracks depend on the corrosion rate.

To sum up, the corrosion expansion of steel bar and cracking of the outer mortar contributed to the fiber strain variations. Thus, it is evident that significant strain variations in the spiral winding fiber reflected concrete cracking. The spiral distributed strain can precisely reflect the corrosion of reinforcement concrete.

## 5. Conclusions

This study developed a new method of monitoring the corrosion of steel bars using spiral wound fibers. Its feasibility was theoretically analyzed and experimentally verified. The main findings of this study are as follows:
(1)The bending curvature of the spiral distributed fiber sensor is related to the winding radius and winding pitch of the sensing fiber. The desired curvature of the winding fiber (0.667 cm^−1^) is a critical parameter for the distributed sensor.(2)Fiber strain variations can precisely reflect the corrosion of reinforcement concrete. Therefore, using spiral distributed fiber sensors to monitor corrosion of steel bars and concrete cracking is a feasible technique.(3)The cracking times and corresponding corrosion rates are closely related to the thickness of the protective layer. Accordingly, the proposed method with the distributed sensor wound on the protective mortar layer is verified to be practical.

## Figures and Tables

**Figure 1 micromachines-12-01451-f001:**
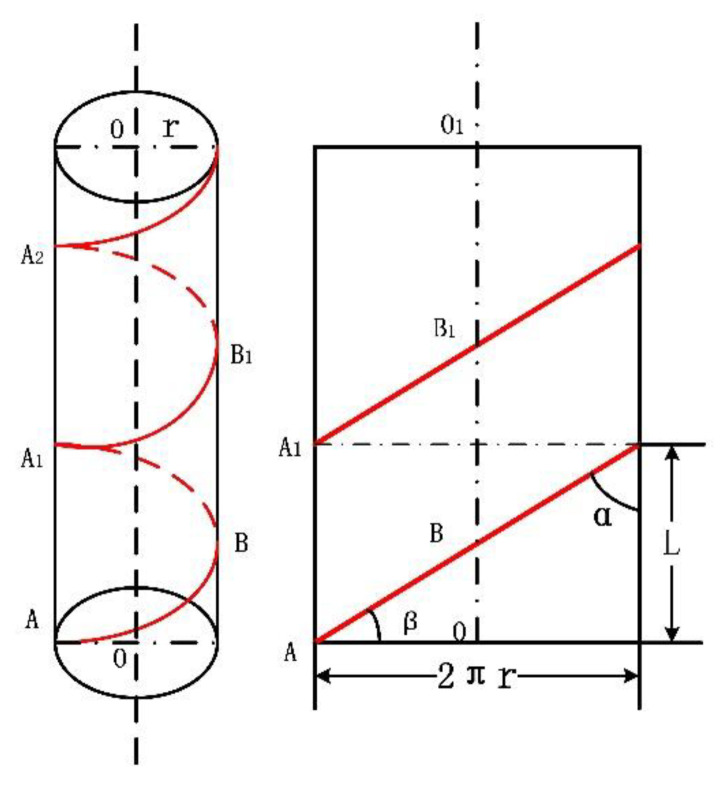
Spiral expansion.

**Figure 2 micromachines-12-01451-f002:**
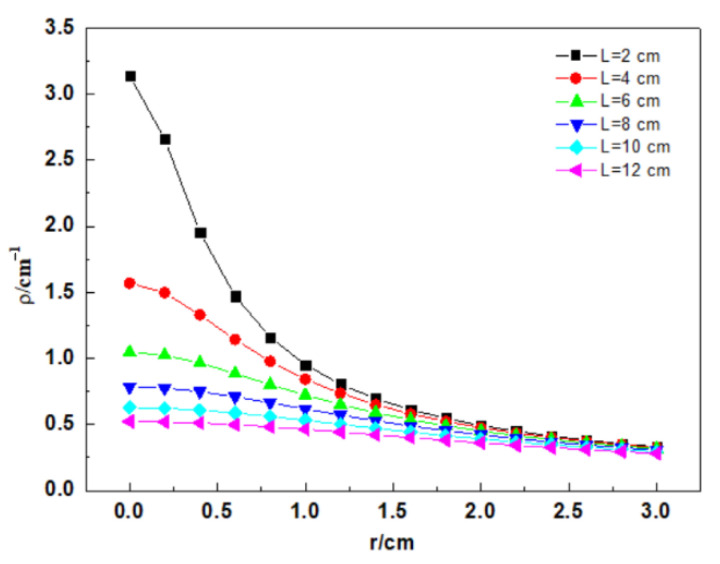
Curvature versus radius of rebar.

**Figure 3 micromachines-12-01451-f003:**
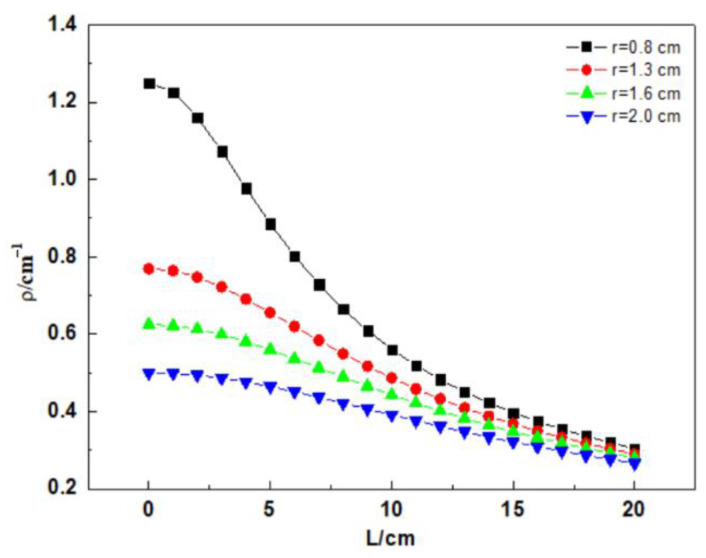
Curvature versus fiber pitch.

**Figure 4 micromachines-12-01451-f004:**
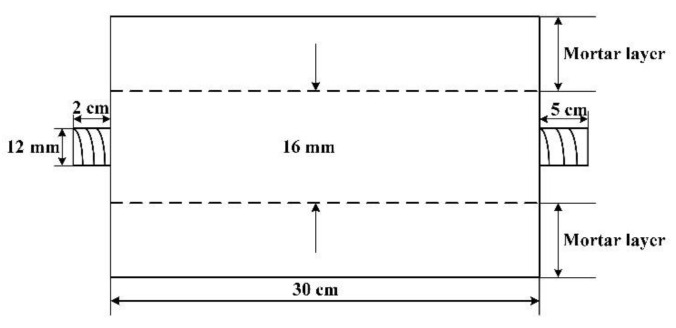
Schematic of the specimen.

**Figure 5 micromachines-12-01451-f005:**
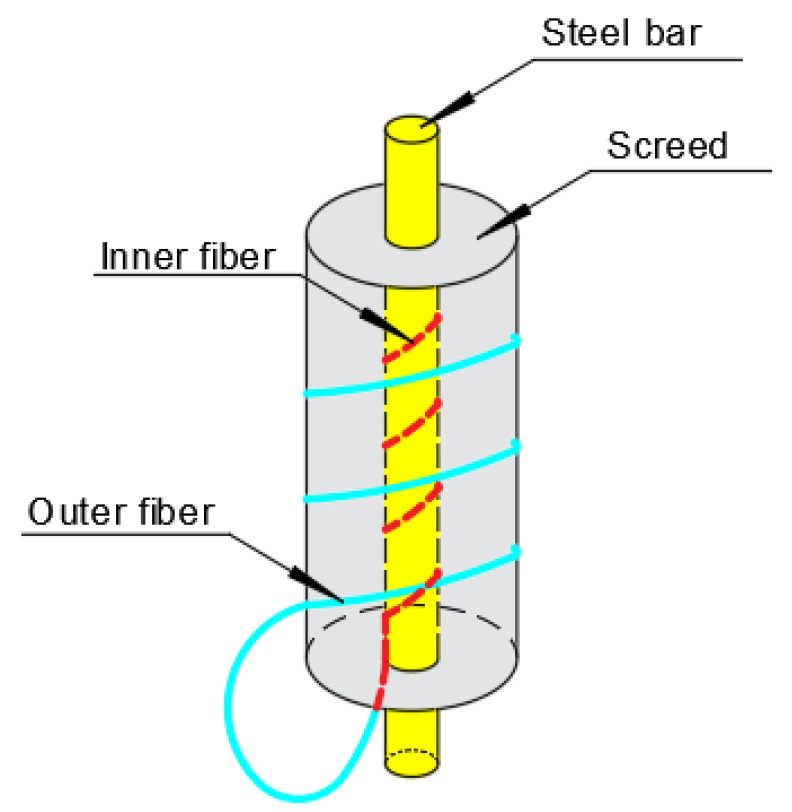
Schematic illustration of the spiral distribution of the optical fiber.

**Figure 6 micromachines-12-01451-f006:**
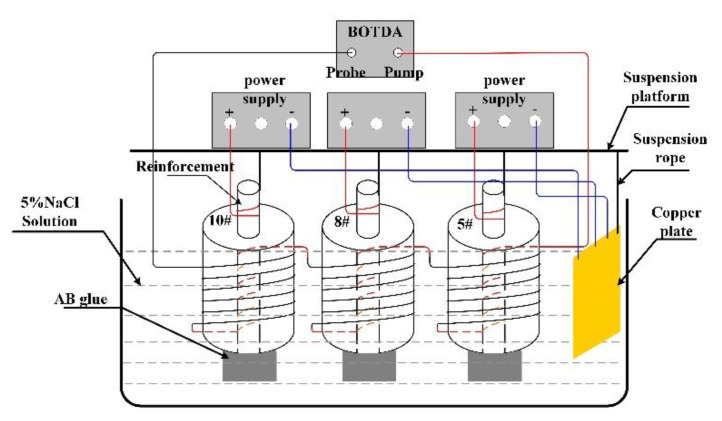
Diagram of accelerated rebar corrosion.

**Figure 7 micromachines-12-01451-f007:**
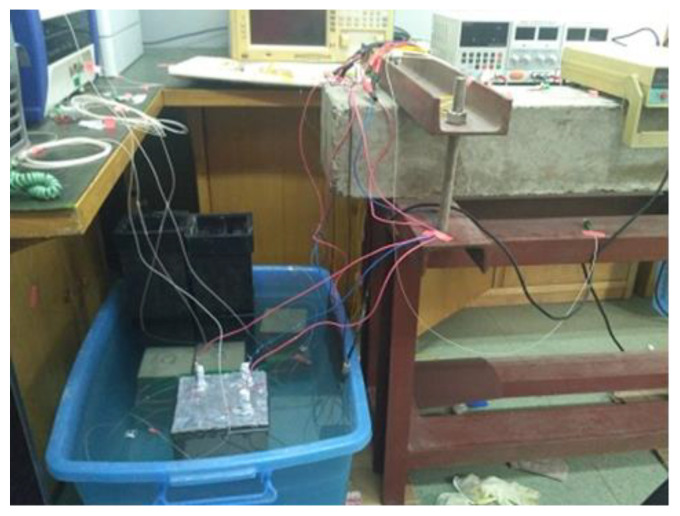
Experimental setup of accelerated corrosion.

**Figure 8 micromachines-12-01451-f008:**
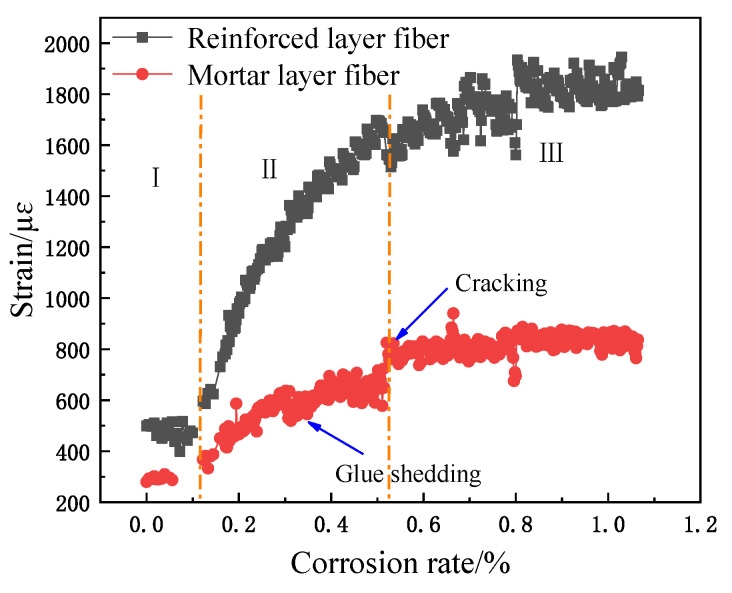
Corrosion curves of sample 5 #.

**Figure 9 micromachines-12-01451-f009:**
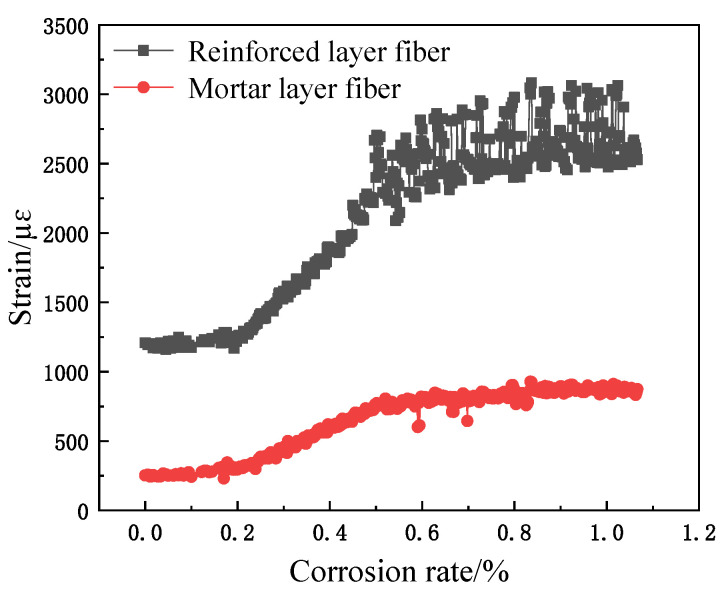
Corrosion curves of sample 8 #.

**Figure 10 micromachines-12-01451-f010:**
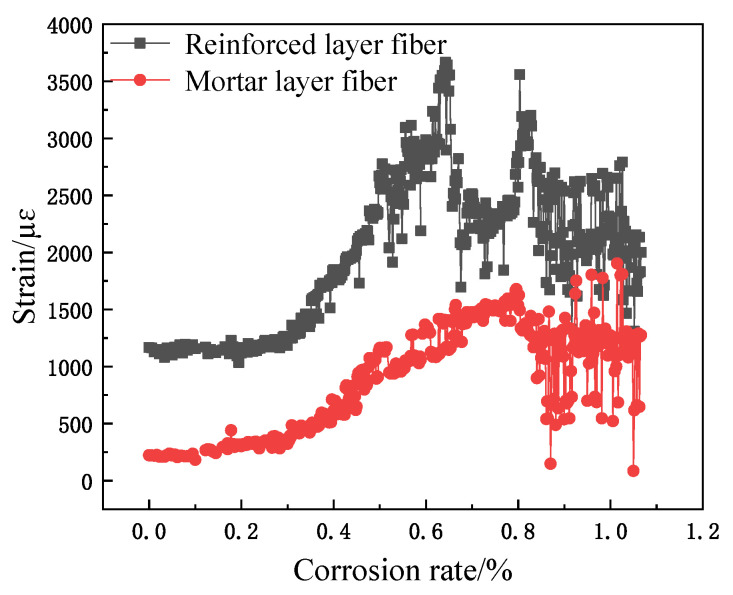
Corrosion curves of sample 10 #.

**Figure 11 micromachines-12-01451-f011:**
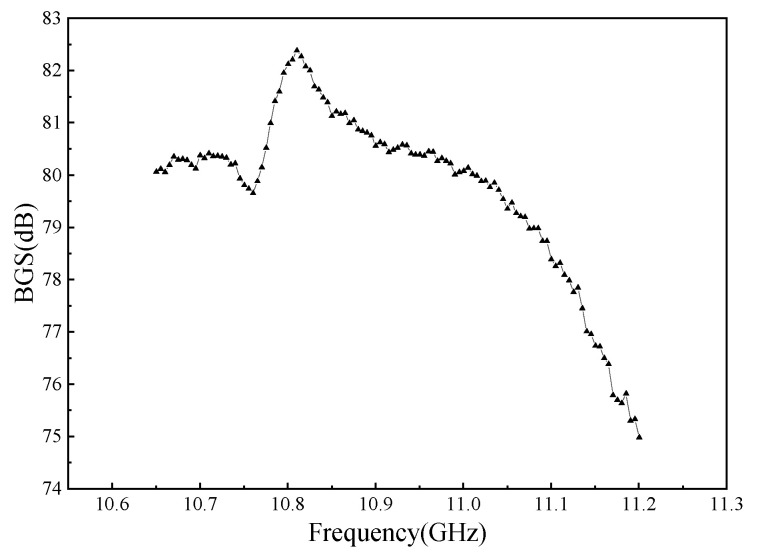
BGS spectrum with a single peak.

**Figure 12 micromachines-12-01451-f012:**
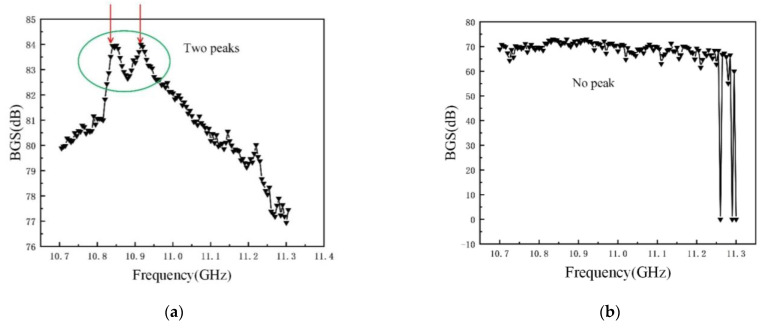
BGS spectrum of the spiral distributed sensor. (**a**) BGS spectrum with double peaks; (**b**) BGS spectrum without Brillouin peak.

**Figure 13 micromachines-12-01451-f013:**
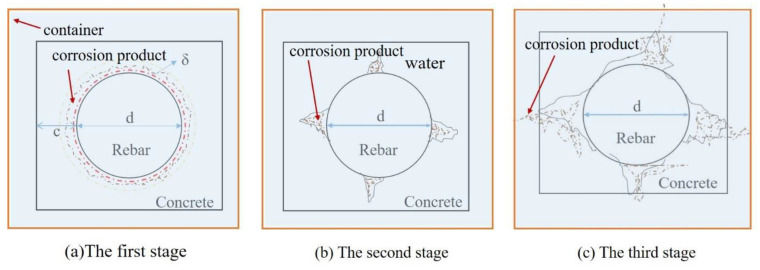
Schematic diagram of the influence of reinforcement corrosion on concrete. (**a**) The first stage; (**b**) the second stage; (**c**) the third stage.

**Figure 14 micromachines-12-01451-f014:**
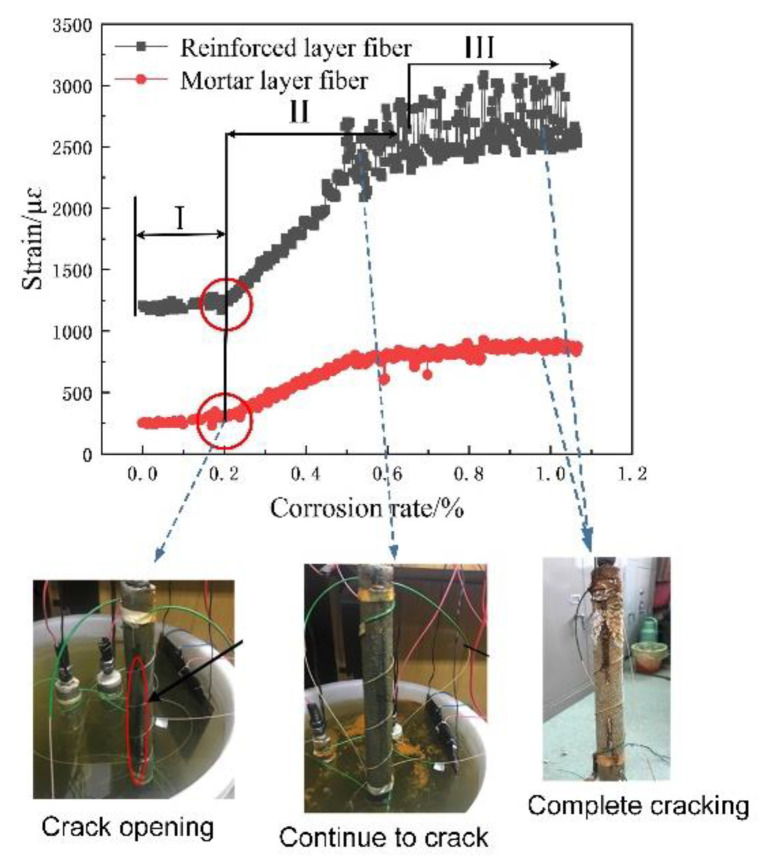
Generated concrete crack.

**Table 1 micromachines-12-01451-t001:** Structural parameters of spiral distributed sensor.

Diameter (mm)	Helix Angle (°)	Winding Pitch (cm)	Bending Curvature (cm^−1^)
16	60	8.7	0.625
16	55	7.1	0.719
16	45	5.0	0.885
16	30	2.9	1.087
26	30	4.7	0.667
32	30	5.8	0.541
36	30	6.5	0.463

**Table 2 micromachines-12-01451-t002:** Corrosion rates and corresponding cracking times.

Specimen	Corrosion Rate (%)	Cracking Time (h)
5#	0.16	6
8#	0.20	10
10#	0.35	16

## Data Availability

The data presented in this study are available from the corresponding author, (J.L.), upon reasonable request.

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
