# Peer review of "A Spiral Distributed Monitoring Method for Steel Rebar Corrosion"

_micromachines, 2021, doi:10.3390/mi12121451_

Round 1
Reviewer 1 Report
In Materials and Methods, lines 5255-53 said that the beta is the complementary angle of beta, that is true? I think the complementary angle is alpha.
The reference [18], the DOI does not correspond to the cited reference.
There are two Subtitles “Materials and Methods”, with different number 2 and 3.
In line 136 is missing the number 9, to include all pictures about the same explanation.
In the part of the double peak on the BGS spectrum, could you explain more detail about it. The authors present 2 figures with and without peaks, in which conditions were measurement these spectrums.
The authors must add a reference or explain what is the reason for the preferable value to the curvature of the winding fiber of 0.625 cm-1.
Author Response
Dear Editor, Dear reviewers
Thank you very much for taking your time to review this manuscript. We truly appreciate all your comments and suggestions. Based on the comments provided in your letter, we have uploaded this letter for reviewer response. Accordingly, we uploaded the revised manuscript with all the changes, the file is MS Word document in "track changes" mode.
Appending this letter is our point-by-point response to the comments raised by the reviewers. Our responses are given directly afterwards for each comment.
We would also like to thank you for allowing us to resubmit a revised manuscript.
We hope that the revised manuscript is accepted for publication in the Micromachines.
Reviewer 1:
Our response: Thank you for pointing out this mistake. We have corrected it in Line 57, as “the complementary angle of ”.
Our response: We have corrected the DOI of reference [18]. And carefully checked the DOI of other references.
Our response: We have revised the Subtitle of Section 3, at Line 119. The Subtitle is changed to “Experiment Program”.
Our response:
In line 136 Figures 8,10,11 is corrected as Figures 8,9,10.
Our response: In general, a single peak in this BGS spectrum is desired, which fail to cause the measurement error. On the contrary, the double peaks in this BGS spectrum and chaotic BGS spectrum, as shown in Figures 12 (a) and (b), cause to the strain fluctuation.
We have added the above discussion to revised manuscript, in Lines 170-173.
Our response: We have added the reference:
“You, S.; Hao, S.; Yin, Z.; Li, X. Analysis and Test of Bend Loss in Single-mode Fiber. Acta Photonica Sinica. 2003, 32, 409-412.”
as reference [26], to explain this preferable value.
Thank you for the comments provided here. We believe we have addressed the comments in our response provided above. Your comments have helped to greatly improve our manuscript and are appreciated.

Reviewer 2 Report
Review comments on a spiral distributed monitoring method for steel rebar corrosion
- Well-structured document and recommended for publication after few corrections. This paper brings out the original findings of the spiral distributed optical fiber-based corrosive monitoring of rebar. The corrosiveness causes the strain on fiber which is explained experimentally and attributed to the light loss ultimately.
- However few corrections are recommended as follows
- Figure 2 and 3 are representing only the numerical relation between the parameters but not the discussing the effect on the light loss. Authors could bring out some discussion on these physical parameters on light loss.
- The relation between the geometric parameters (physical length of rebar, number of wounds) was not discussed in detail with respect to optical light loss. The temporal light loss graph may enhance the understanding of reader while the corrosion is taking place.
- The relevance of figure 11 in this manuscript is not clear in discussions
Some general comments
- Authors should compare their results with the published data and different results.
- How their results will be affected if they include energy loss in the fiber itself.
- Finally, I recommend that the paper should be revised taking care of the above comments. I wish to resend this paper after corrections and revise my comments.
Author Response
Reviewer 2:
Our response:
Reference 26 has investigated the function between bend loss and bend radius. While this manuscript discusses the effect of radius of rebar and fiber pitch on the curvature of a spirally distributed sensor based on the results of Reference 26. Reciprocal of the bending curvature is the bending radius of a spirally distributed sensor. As a result, the bending curvature in turn affects the bend loss of fiber. Eventually, the radius of rebar and fiber pitch affect the bend loss. The related discussion has been added to the revised manuscript, in section 2.1, Lines 63-67.
Our response:
To decrease the light loss induced by fiber bend, the minimal bending radius is achieved in this study. Number of wounds has no effect on the light loss when winding radius exceed the minimal bending radius based on the reference 26. The light loss is measured according to the optical fiber bending measurement method. The bend loss is not achieved while the corrosion is taking place. The related discussion has been added to the revised manuscript, in section 2.1, Lines 63-67.
Our response: We added some references to illustrate the different stages of corrosion process in Section 1. It can be seen that the experiment results of our manuscript are similar to the corrosion process of other papers. we added the discussion in Lines 152-154 to further support the experiment results.
Thank you for the comments provided here. We believe we have addressed the comments in our response provided above. Your comments have helped to greatly improve our manuscript and are appreciated.

Round 2
Reviewer 2 Report
The revised manuscript is good. It can be accepted.